



# Characterization of the benthic biogeochemical dynamics after flood events in the Rhône River prodelta: A data-model approach

Eva Ferreira[1],[∗], Stanley Nmor[1],[∗], Eric Viollier[1], Bruno Lansard[1], Bruno Bombled[1], Edouard Regnier[1], Gaël Monvoisin[2], Christian Grenz[3], Pieter van Beek[4], and Christophe Rabouille[1]

[1]Laboratoire des Sciences du Climat et de l'Environnement, LSCE/IPSL, CEA-CNRS-UVSQ-Université Paris-Saclay, 91198, Gif sur Yvette, France
[2]Géosciences Paris-Saclay (GEOPS), CNRS and Université Paris-Saclay, 91405, Orsay, France
[3]Mediterranean Institute of Oceanography (MIO), Aix Marseille Université, Université de Toulon, CNRS, IRD, MIO UM 110, 13288, Marseille, France
[4]Laboratoire d'Etudes en Géophysique et Océanographie Spatiales (LEGOS), CNES/CNRS/IRD/Université Toulouse III Paul Sabatier, 31400, Toulouse, France

[∗]: The two first authors (E. Ferreira and S. Nmor) contributed equally to this paper.

Correspondence to: Eva Ferreira (eva.ferreira@lsce.ipsl.fr)

**Abstract.**

At the land-sea interface, the benthic carbon cycle is strongly influenced by the export of terrigenous particulate material across the river-ocean continuum. Episodic flood events delivering massive sedimentary materials can occur, but their short-term impact on carbon cycling is poorly understood. In this paper, we use a coupled data-model approach to estimate the temporal variations of sediment-water fluxes, biogeochemical pathways and their reaction rates during these abrupt phenomena. We studied one episodic depositional event in the vicinity of the Rhône River mouth (NW Mediterranean Sea) during the fall-winter of 2021-2022. The distribution of dissolved inorganic carbon (DIC), sulfate ($SO_4^{2-}$) and methane ($CH_4$) were measured in sediment porewater collected every 2 weeks before and after the deposition of a 25 cm sediment layer during the main winter flood event. Significant changes in the distribution of DIC, $SO_4^{2-}$ and $CH_4$, concentrations were observed in the sediment porewaters. The use of an early diagenetic model (FESDIA) to calculate biogeochemical reaction rates and fluxes revealed that this type of flooding event can increase the total organic carbon mineralization rate in the sediment by 75% a few days after deposition, essentially by increasing the sulfate reduction contribution to total mineralization relative to non-flood depositional period. It predicts a short-term decrease of the DIC flux out of the sediment from 100 to 55 $mmol\ m^{-2}\ d^{-1}$ after the deposition of the new sediment layer with a longer-term increase by 4%, therefore implying an initial internal storage of DIC in the newly deposited layer and a slow release over relaxation of the system. Furthermore, examination of the stoichiometric ratios of DIC and $SO_4^{2-}$ as well as model output over this five-months window shows a decoupling between the two modes of sulfate reduction following the deposition - organoclastic sulfate reduction (OSR) intensified in the newly deposited layer below the sediment surface, whereas anaerobic oxidation of methane (AOM) intensified at depth below the former buried surface. This depth-wise bifurcation of both pathways of sulfate reduction in the sediment column is clearly related to the deepening of the sulfate-methane transition zone (SMTZ) by 25 cm after the flood deposition. Our findings highlight the significance of short-term transient biogeochemical processes at the seafloor and provide new insights on the benthic carbon cycle in the coastal ocean.



## 1 Introduction

River-dominated ocean margins (RiOMars) are crucial areas linking land and open ocean. They play a key role in marine nutrient and carbon cycles (McKee et al. 2004; Cai 2011; Bauer et al. 2013; Bianchi et al. 2018). These dynamic environments are known to have high riverine input and sedimentation rate (Aller 1998). Furthermore, coastal sediments

account for 85% of long-term organic carbon burial in the ocean, with deltaic environments accounting for the majority (Burdige 2005), but they are also powerful biogeochemical reactors (Aller et al. 1996; Rassmann et al. 2016). The large deposition of riverine (or terrigenous) particulate organic matter (POM) on the seafloor can result in the storage of organic carbon (OC) but also in strong benthic mineralization rates, dominated by sulfate reduction and methanogenesis (Mucci et al. 2000; Burdige and Komada 2011). In deltaic sediments, which receive large amounts of POM, anaerobic respiration is one

of the most important pathways for the remineralization of organic carbon (Canfield 2004; Canfield and Thamdrup 2009; Pastor et al. 2011a). AS an example, the prominent anoxic pathway in the Rhône River prodelta is sulfate reduction accounting for approximately 70% of the total organic carbon mineralization rate in these sediments (Pastor et al. 2011a). This anoxic mineralization of organic carbon is supplemented by methanogenesis which can account for up to 35% of total organic matter degradation in sediment where a portion of reactive organic matter remains after complete sulfate exhaustion

(Egger et al. 2016). The methane fluxes are controlled by the anaerobic oxidation of methane (AOM) in the subsurface sulfate-methane transition zone (STMZ; Boetius et al., 2000). Together, these processes modulate anoxic-based carbon cycling in coastal and deltaic sediments, therefore generating large quantities of dissolved inorganic carbon (DIC), and RiOMar systems are often considered as $CO_2$ sources to the atmosphere (Cai 2011; Bauer et al. 2013).

Flood events are especially significant along river-dominated margins and particularly for smaller river systems where

sediment transport to the ocean preferentially occurs during extreme precipitation events (Bourrin et al. 2008; Lee et al. 2015). These materials can be subjected to secondary transport by waves and currents with a repeated cycle of resuspension and deposition (Ulses et al., 2008; Moriarty et al. 2017) as they discharge to the adjacent deltas and shelves. Furthermore, episodic events are also important in determining the locations and magnitude of hotspots OC burial on the coastal margin. This is especially true during large storms that can greatly increase both river discharge and sediment load, resulting in

increased sediment discharge to depositional zones along the shelf (Eglinton 2008). During flood periods, large amounts of sediment and terrigenous OM are delivered to the adjacent delta and shelf. For example, the Eel River (Northern California) is a major source-to-sink conduit for large sediment transport, delivering between 60 and 80% of discharged fine-grained sediment to the adjacent marine depocenter during large winter storms (Wheatcroft and Sommerfield 2005). Similar large deposition of sediment over relatively short period of time have been documented elsewhere, near the Island of Taiwan (Liu

et al., 2013), the Mekong delta (Manh et al. 2014) or the Yangtze River-estuary depositional system (Dai et al. 2018) to name a few.

In the Rhône River (NW Mediterranean Sea), flood events can account for 80% of annual terrigenous particle inputs (Antonelli et al. 2008; Eyrolle et al. 2012), which at times can deliver up to 30 cm of sediment deposition on the Rhône River prodelta located in the Gulf of Lion (Charmasson et al. 1998; Antonelli et al. 2008; Lansard et al. 2009; Cathalot et al.

2010). These sediments are mostly deposited in the prodelta as previously shown by Wu et al. (2018) using beryllium-7 ([7]Be), a natural short-life radionuclide which traces deposits of riverine suspended particulate matter (SPM). These winter events are abrupt and therefore difficult to document precisely. As a result, few studies have been conducted on the biogeochemical response of coastal sediment following intense export of sediment and organic carbon (Cathalot et al. 2010; Pastor et al. 2018). Furthermore, we can expect that the frequency and intensity of flood events will increase as a result of

climate change (Day et al. 2019; Lionello et al. 2023). However, due to the unpredictable nature of meteorological and flood events, it is difficult to monitor these intense events.



Many efforts have been made to incorporate biogeochemical processes operating in the sediment into mathematical models (Berner 1980). These early diagenetic models have been heavily used to investigate the fate and transport of a selected set of chemical species in the seafloor (Aguilera et al. 2005). Recent non-steady state diagenetic models based on previous

numerical representations of sediment transport and reactions (Rabouille and Gaillard 1991; Soetaert et al. 1996; Wang and Van Cappellen 1996; Berg et al. 2003) have demonstrated the importance of explicitly depicting event-driven processes (De Borger et al. 2021; Nmor et al. 2022). The benefit of these models is that they take deposition thickness into consideration as a vital parameter for reproducing such an episodic event (Nmor et al. 2022). Combining sediment and porewater data can help to constrain model inputs and aid in the simulation of such depositional events.

The goal of this study is to examine the transient evolution of benthic carbon mineralization processes and their impact on sediment-water exchange during a flood event marked by large sediment deposition. We intend to characterize and quantify the changes that occur on several biogeochemical pathways and fluxes during these periods of substantial deposition of sedimentary material. We use a dualistic approach to solve this question by combining bi-monthly observational data on sediment evolution with a non-steady state early diagenetic model that calculates biogeochemical rates and fluxes. This

multivariate perspective gives us a better understanding of the factors that control organic carbon remineralization and the relative shares of organoclastic sulfate reduction (OSR) and methanogenesis, together with anaerobic oxidation of methane (AOM). We therefore address the question of how massive material deposition affects the biogeochemical carbon cycle in coastal sediment.

## 2 Materials and Methods

### 2.1 Study site

The Rhône River is the main source of freshwater, nutrients, organic matter and sediment for the Mediterranean Sea (Madron et al. 2000). It is characterized by a drainage basin of 97800 $km^2$ and an average water discharge of 1700 $m^3\ s^{-1}$ with a marked seasonality between low-water discharge (<700 $m^3\ s^{-1}$) in summer and high-water discharge (>3000 $m^3\ s^{-1}$) in fall and winter (Pont et al. 2002). The Rhône River turbidity plume extends mainly southwestward into the Gulf

of Lion, with an average thickness of 1 m (up to 5 m) (Many et al. 2018). The Gulf of Lion is a microtidal, wave-dominated system, with a tidal range of 30 to 50 cm. Due to salt induced flocculation (Thill et al. 2001), most suspended particulate matters (SPM) carried out by the Rhône River settle in front of the mouth, on the prodelta (Maillet et al. 2006; Estournel et al. 2023). The study site (station Z, water depth 20 m, Fig. 1) is located on the delta front, at a distance of 2 km from the river mouth, and is characterized by a mean apparent accumulation rate of up to 35 to 48 $cm\ yr^{-1}$ (Charmasson et al. 1998).

The site is defined by geographic coordinates (lat. 43°19.1'N and long. 04°52.0'E), but the constraints of sea work (e.g. ship drift) lead to a positioning variability estimated at a perimeter of 60m around these coordinates.



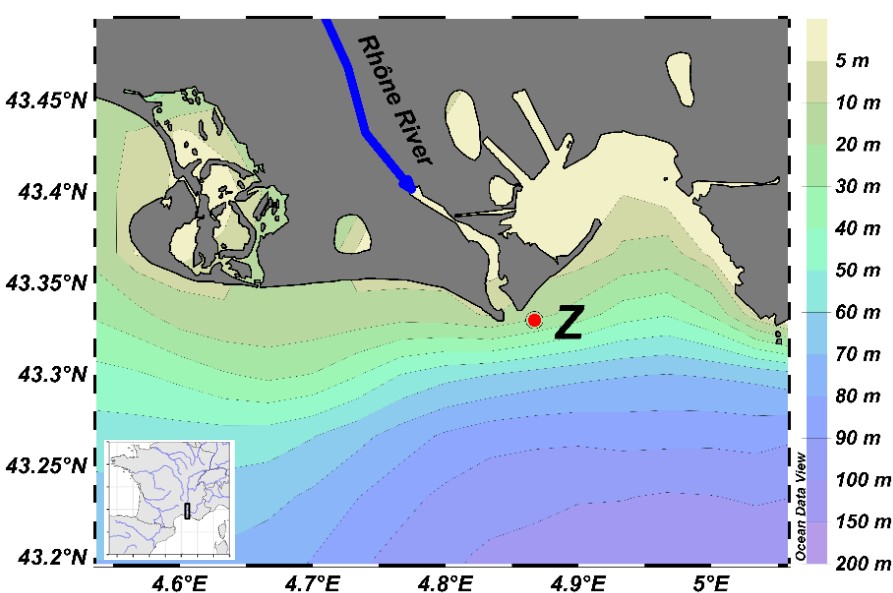

Figure 1: Map of the Gulf of Lion (Rhône prodelta) including the location of the sampling station (station Z).

Table 1: Temporal sampling coverage and location of sampling sites during the winter season of 2021-2022.

| Cruises | Date | Rhône River water flow (m3 s-1) | Lat (°N) | Lon (°E) |
|---------|------|----------------------------------|----------|----------|
| SB7 | 03 November 2021 | 2057 | 43°19.066' | 4°52.023' |
| SB7bis | 19 November 2021 | 830 | 43°19.066' | 4°52.023' |
| SB8 | 01 December 2021 | 905 | 43°19.032' | 4°51.952' |
| SB9 | 07 January 2022 | 2533 | 43°19.111' | 4°52.048' |
| SB9bis | 19 January 2022 | 1318 | 43°19.096' | 4°52.034' |
| SB10bis | 23 February 2022 | 1972 | 43°19.131' | 4°52.071' |
| SB11 | 08 March 2022 | 1110 | 43°19.108' | 4°52.089' |


The fall-winter monitoring (AMOR SB) took place bi-monthly from November 2021 to March 2022 (Table 1) when the weather permitted with sampling cruises onboard the *RV Antédon II* (IFREMER-FoF). The Rhône River flows were recovered from the Hydroréel database at the Tarascon-Beaucaire station (hydrometric station V720001002). The SPM content are recovered from the database of the Rhône Sediments Observatory. Missing data are estimated empirically using

the relationship between flows and Cs determined from sediment rating curves (Lepage et al. 2022).

### 2.2 Sediment and porewater sampling and analyses

Sediment cores were collected at each of the cruises reported in Table 1 with an UWITEC single corer (USC 09000) equipped with a weight of 30 kg. The length of the coring tubes was 120 cm with an internal diameter of 9 cm. At least, two sediment cores were retrieved with a well-preserved sediment-water interface (SWI). One core was dedicated to the

sampling of sediment porewaters and the second core was cut into slices for further laboratory analysis. Sediment porewaters were extracted using syringes connected to porous Rhizon with an average pore diameter of 0.1 µm (Seeberg-Elverfeldt et al.





2005). The vertical sampling resolution was 2 cm for the first 10 cm and then 4 cm down to the end of the core. At each sampled depth between 12 and 15 mL of porewater were extracted. The content of each syringe was immediately subsampled after extraction. For dissolved inorganic carbon (DIC) analysis, 5 mL samples were poisoned with 10 μL of

supersaturated $HgCl_2$ and stored in 10 mL glass vials at 4°C until analysis. Concentrations of DIC were analyzed by a LI-COR infrared detector with a DIC analyzer ASC-1 (Apollo SciTech) on 0.75 mL samples, as described in Rassmann et al. (2016). The relative uncertainty was ± 0.2%. For $SO_4^{2-}$ analysis, 2 mL were subsampled and acidified with 8 μL of HNO3 and stored in Eppendorf tubes at 4°C until analysis. Concentrations of $SO_4^{2-}$ were analyzed on 100 μL samples with a liquid phase ion chromatography system (ICS 1000 $Dionex^{TM}$) with AG14 precolumn, AS14 column and AERS 500 suppressor

configuration at Geosciences Paris-Saclay laboratory, as described in Rassmann et al. (2020). The relative uncertainty was ± 0.3%.

On the same core, sub-core samples were taken for $CH_4$ with a 5 cm resolution in side holes using two sharpened 5 mL syringes of 1 cm diameter. The contents of two syringes of the same level were quickly introduced into empty pre-weighed 60 mL vials with 35 mL of KOH (1 $mol\ L^{-1}$). The vials were then directly sealed, shaken and stored upside down in the

dark. Back in the laboratory, $CH_4$ concentrations were analyzed with a micro-gas chromatograph Agilent Technologies® 490 GC. Measurements were made in three 60-second analyses, with 1.5 ml gas samples taken from the "headspace" (Rassmann et al., 2020). The calibration was performed with a standard gas of $CH_4$ at 104 ppm with a reproducibility of 1%. Data obtained indicate a percentage of $CH_4$ in the headspace which allows the calculation of $CH_4$ quantity in headspace by dividing the volume of $CH_4$ by the molar volume of a gas at 1 atm. This quantity is then used to back calculate the $CH_4$

concentration in the porewater using porosity and sediment weight, with an estimated accuracy of 5%.

The second sediment core was sliced as follows: every 0.5 cm for the first 2 cm of the core, every 1 cm down to 10 cm, every 2 cm down to 20 cm and finally every 5 cm for the rest of the core. Sediment samples were stored in freezer bags preserved at -20 °C. One part of the sediment samples was used to determine the organic carbon (OC) content, reported in % dry weight sediment. Sediment layers were freeze-dried, crushed and decarbonated by successive acidification baths (HCl

1%) over several days after rinsing. Homogenized and accurately weighed subsamples were analyzed by a Carlo-Erba NA-1500 Elemental Analyzer. The average OC contents are calculated for the first 30 cm sediment before and after the flood.

Another part of this sediment core was used to analyze beryllium-7 ([7]Be) activity within three months after sample collection by using low-background gamma-ray spectrometry at the LAFARA underground laboratory (van Beek et al. 2013). Between 8.0 and 12.5 g of dry sediment were analyzed during 48 h using a Mirion-Canberra planar detector (germanium crystal of

230 cm[3]) equipped with LYNX (Mirion-Canberra) electronics and an electric cooling system (CryoPulse® 5 plus provided by Mirion-Canberra). The 7Be activities are reported with 2 sigma uncertainties.

### 2.3 Stoichiometric ratio

The anaerobic mineralization of buried organic matter by sulfate reduction (Reaction R1) and the anaerobic methane oxidation (AOM, Reaction R2) reactions provide theoretical, stoichiometric ratios ($r_{c:s}$) of $SO_4^{2-}$ consumption to bicarbonate

ion production.

$$2CH_2O + SO_4^{2-} \rightarrow 2HCO_3^- + H_2S \quad (R1)$$
$$CH_4 + SO_4^{2-} \rightarrow HCO_3^- + HS^- + H_2O \quad (R2)$$

This ratio can be used to identify the key process that dominates mineralization in sediments from porewater measurements (Burdige and Komada 2011). The $r_{c:s}$ (Equation 1) of the present dataset were calculated as described by Burdige and Komada (2011). The slope of the property-property plot of $\Delta DIC$ versus $\Delta SO_4^{2-}$ was corrected by the diffusion coefficient





ratio at in situ temperature (Li and Gregory 1974) in order to eliminate the effect of transport by diffusion (Burdige and Komada 2011).

$$r_{C:S} = \frac{D_{HCO_3^-}}{D_{SO_4^{2-}}} \cdot \frac{\Delta DIC}{\Delta SO_4^{2-}} \quad (1)$$

Before the flood event $r_{C:S}$ were calculated on the whole core. After the deposition event, two $r_{C:S}$ were calculated as a function of depth, the first on a surface layer between 1 and 25 cm and the second from 25 cm to the bottom of the core.

**2.4 Numerical modelling**

The model FESDIA is a time-dependent early diagenesis model designed for perturbation studies. This model is made up of a set of coupled nonlinear partial differential equations that describe the distribution of porewater species at different depths. This model is notable for its ability to simulate event-driven dynamics such as sudden sediment deposition as a result of flood input. Details of the model formulations and equations were described in Nmor et al. (2022). Here, we briefly outline

important processes involving sulfur and methane cycle as well as parameterization considered necessary for the representation of the winter flood situation the Rhône River prodelta.

The model considers the entire sequence of OM remineralization pathways in the sediment, including OM remineralization coupled to oxygen, nitrogen, iron and manganese oxides, sulfate, and, finally, methane production. In general, organic matter oxidation follows the formalism of a cascading sequence of these terminal electron acceptors (Canfield and Thamdrup,

2009). The organic matter modelled is made up of two degradable fractions with different reactivities. This decay is modelled as a first-order rate law and is dependent on the limitations of specific oxidants and their inhibition. Secondary reactions involving reduced species include nitrification, reoxidation of reduced metals, methane oxidation (see below) via oxygen, sulfide reoxidation by iron and manganese hydr(oxides), and iron-sulfide precipitation. Table 2 contains a summary of the parameters used in the model. These values were either derived from previous steady-state modeling works in the

Rhône prodelta sediment (Pastor et al. 2011a; Ait Ballagh et al. 2021) or in other cases, where no model parameter value was known for the Rhône prodelta sediment, values from other literature sources were calibrated with the observed data.

**Table 2: Summary of parameters used in the FESDIA model. (I) independent parameters derived from experiment or field observation external to actual data being simulated (C) constrained parameters obtained from range of literature sources (M) model-derived parameters fitted to the observed data. Literature sources includes (1) Pastor et al. (2011a), (2) Soetaert et al.**

**(1996), (3) Ait Ballagh et al. (2021), (4) Rassmann et al. (2020), (5) Wang and Van Cappellen (1996) and (6) Van Cappellen and Gaillard (2018).**

|  | Description | Parameters | Units | Type | Source |
|---|---|---|---|---|---|
| Cflux | total organic C deposition | 15000 | nmolC cm-2 d-1 | I | 1 |
| pFast | part FDET in carbon flux | 0.5 | - | C | 1 |
| FeOH3flux | deposition rate of FeOH3 | 7000 | nmolC cm-2 d-1 | M | - |
| MnO2flux | Flux of Mn Oxides | 1500 | nmolC cm-2 d-1 | M/C | -/5 |
| NCrFdet | NC ratio FDET | 0.14 | molN/molC | I | 2 |
| NCrSdet | NC ratio SDET | 0.1 | molN/molC | I | 2 |
| O2bw | upper boundary O2 | 238 | mmol m-3 | M | - |
| NO3bw | upper boundary NO3 | 0 | mmol m-3 | M | - |
| NH3bw | upper boundary NH3 | 0 | mmol m-3 | M | - |



| | Description | Parameters | Units | Type | Source |
|---|---|---|---|---|---|
| CH4bw | upper boundary CH4 | 0 | mmol m-3 | M | - |
| DICbw | upper boundary DIC | 2360 | mmol m-3 | M | - |
| Febw | upper boundary Fe2+ | 0 | mmol m-3 | M | - |
| H2Sbw | upper boundary H2S | 0 | mmol m-3 | M | - |
| SO4bw | upper boundary SO42- | 33246 | mmol m-3 | M | - |
| Mnbw | upper boundary Mn2+ | 0 | mmol m-3 | M | - |
| w | advection rate | 0.08 | cm d-1 | M | - |
| por0 | surface porosity | 0.83 | - | I | 4 |
| pordeep | deep porosity | 0.65 | - | I | 1/4 |
| porcoeff | porosity decay coefficient | 2 | cm | I | 1/4 |
| biot | bioturbation coefficient | 0.05 | cm2 d-1 | C | 1 |
| biotdepth | depth of sediment mixed layer | 5 | cm | I | 3 |
| biotatt | attenuation coeff below biotdepth | 1 | cm | I | 3 |
| irr | bio-irrigation rate constant | 0.3 | d-1 | M | - |
| irrdepth | depth of irrigated layer | 7 | cm | I | 3 |
| irratt | attenuation coeff below irrdepth | 1 | cm | I | 3 |
| temperature | temperature | 15.6 | dgC | M | - |
| salinity | salinity | 37.8 | psu | M | - |
| TOC0 | refractory Carbon conc | 1.0 | % | M | - |
| rFast | decay rate FDET | 0.5 | d-1 | C | 1 |
| rSlow | decay rate SDET | 0.0031 | d-1 | C | 1 |
| rH2SMnox | Rate of Reoxidation of H2S by MnOx | 0.001728 | cm3 nmol-1 d-1 | M/C | -/5 |
| rFeS | rate constant of FeS production | 0.5 | cm3 nmol-1 d-1 | I | 5 |
| rMnFe | Rate constant of Fe Reoxidation with MnOx | 6.48e-06 | cm3 nmol-1 d-1 | M/C | -/5 |
| RAOM | Rate constant for AOM | 3 10-5 | cm3 nmol-1 d-1 | C | 6 |
| alphaFDET | Enrichment factor for FDET | 4 | - | M | - |
| alphaSDET | Enrichment factor for SDET | 1.8 | - | M | - |
| alphaFeOH3A | Enrichm. factor for FeOH3A | 1 | - | M | - |
| alphaFeOHB | Enrichm. factor for FeOHB | 1 | - | M | - |
| alphaMnO2A | Enrichm. factor for MnO2A | 1 | - | M | - |
| alphaMnO2B | Enrichm. factor for MnO2B | 1 | - | M | - |





### 2.5 Methanogenesis

Below the sulphidic zone, organic carbon that remains is subsequently remineralized via methanogenesis. The product of this
fermentation of organic matter in depth by anaerobic archaea is methane ($CH_4$) and can be represented as follows Reaction
3 :

$$2\,CH_2O \rightarrow CH_4 + CO_2 \qquad (R3)$$

In the Rhône River proximal prodelta, evidence of high apparent sedimentation deposition (>30 $cm\,yr^{-1}$) and high
particulate organic carbon flux (657 $gC\,m^{-2}\,yr^{-1}$) have been observed (Madron et al. 2000; Pastor et al. 2011b). As a
result, high methane production in deeper sediment is likely (Garcia-Garcia et al. 2006; Pozzato et al. 2018; Rassmann et al.
2020). In the model, the accumulation of methane derived from carbon remineralization is limited by the equilibrium
between dissolved and free gas which can occurs at around 90 ppm (or 6 mM) in shallow sediment of the Rhône prodelta
(Garcia-Garcia et al. 2006). This is done by considering methane removal into either free gas as Equation 2:

$$CH_{4gas} = max\left(0, k_{gas} * \left(CH_4 - CH_{4equil}\right)\right) \qquad (2)$$

where $CH_{4equil}$ is the equilibrium concentration for which observed/simulated methane transition to hydrate or gas phases.

### 2.6 Methane oxidation

The methane produced deep down in the sediment can diffuse upward and be re-oxidized in the presence of oxygen
(Reaction 4) with a simple first-order rate expression used in the model:

$$CH_4 + 2O_2 \rightarrow CO_2 + 2H_2O \qquad (R4)$$

However, an important part of this investigation involves the interaction between the sulfur and methane cycle. Critical to
this link is the role of anaerobic oxidation of methane (AOM) (Reaction 2). The AOM is a vital microbial process which acts
as a barrier to the extent of the upward methane flux from the deeper sediment. The AOM occurs at the nexus of sulfate
depletion and methane production; at a depth horizon typically referred to the sulfate-methane transition zone (SMTZ). This
reaction is modelled as a first-order process involving both $CH_4$ and $SO_4^{2-}$.

$$AOM = R_{AOM} \times CH_4 \times SO_4^{2-} \qquad (3)$$

where $R_{AOM}$ is the constant apparent AOM reaction rate and as this pathway of sulfate reduction occurs at a much slower rate
than the sulfate reduction coupled to organic carbon oxidation ( Van Cappellen and Gaillard 2018), the value $R_{AOM}$ was set to
$3 \times 10^{-5}$ cm3/nmol/d.

### 2.7 Model configuration

The FESDIA model was implemented in a 1 m sediment domain with variable depth resolution. Sediment thickness
increases from 1 mm at the surface to about 6 cm at the base of the domain. For our application, we used a sedimentation
rate of 30 $cm\,yr^{-1}$ (Lansard et al. 2009) and the degradation constant of the labile carbon ($rFast$) was tuned to 0.05 $d^{-1}$.
Other parameters relevant for this particular simulation were derived from other literature sources and a listing is provided in
Table 2.

Porosity was modelled as an exponential decay with depth increasing from 0.83 at the surface to an asymptotic value of 0.65
at depth following the obtained data. Bioturbation was constant in the first 5 cm with rate of 0.05 $cm^2\,d^{-1}$ and exponentially
attenuated below with reduced fauna activity. Based on the low bioturbation rate observed at station Z and the dominance of



flood deposition on sedimentation (Pastor et al., 2011), the FESDIA model is used with a constant bioturbation rate over the study period (Nmor et al., 2022). Solutes pumping via bio-irrigation was also modelled. A summary of the parameters used in the model is described in Table 2.


The deposition of flood materials was carried out in similar manner as described in Nmor et al. (2022). Here, we imposed a singular flood scenario with a thickness of 25 cm. The inclusion of this single event was dictated by the dominant presence of an abnormally high SPM concentration observed during the winter flooding season as recorded by the SORA monitoring station located in Arles, 40 km upstream from the river mouth (Fig. 2). As such, we assumed that deposition during this flood period only lagged by a few days from the observance of high TSM load. This forces the date used for the deposition in the model (03 January 2022). The deposited material thickness in the model is indirectly diagnosed using measurement of porewater solute distribution and strengthened by beryllium-7 data collected after particle settling (see Section 3).


As described in Nmor et al. (2022), the deposited flood layer can have a different particulate composition than the pre-existing sediment. Depending on the nature of the flood, it can be enriched or depleted in reactive compounds (two pools of organic matter ($C_{fast}^{org}$, $C_{org}^{slow}$), of manganese ($MnO_{2A}$ and $MnO_{2B}$) and amorphous iron ($FeOOH_A$ and $FeOOH_B$)) which is translated in the model by the enrichment factor (α). This α factor was set to the values reported in Table 2.


## 3 Results

### 3.1 Water discharge and SPM concentrations in the Rhône River

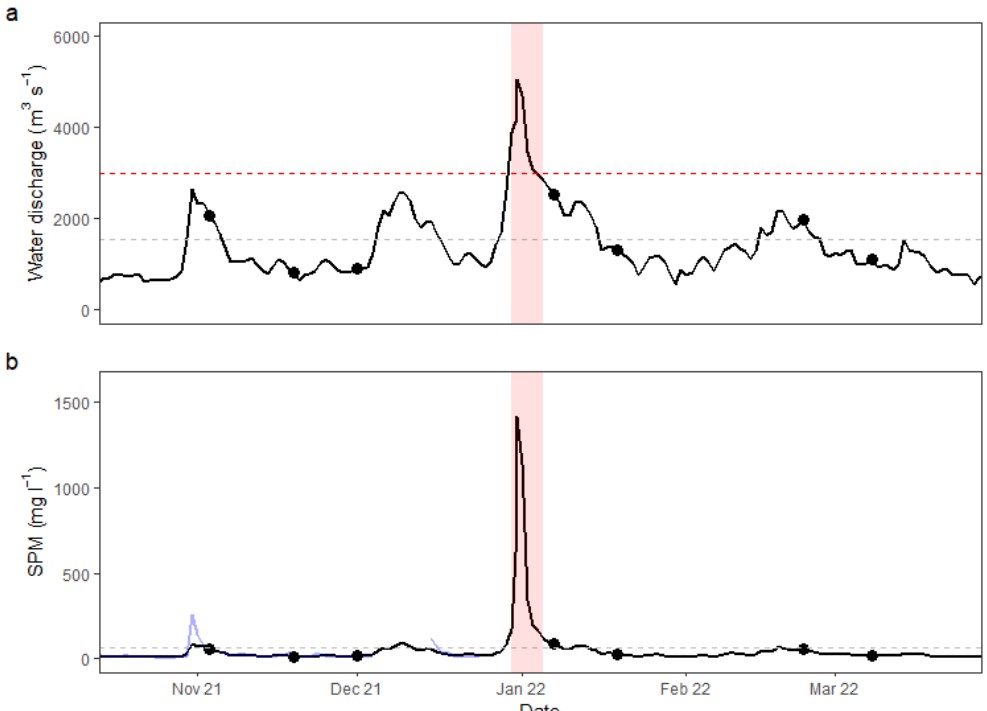

**Figure 2: (a) Mean daily water discharge of the Rhône River at Beaucaire-Tarascon, located 60 km upstream the river mouth. The grey dotted line symbolizes the average water discharge level. The red dotted line symbolizes the flood level at Beaucaire-Tarascon with the flood period symbolized with the red bar. (b) Total suspended particulate matter (SPM) concentration in the Rhône River**



at SORA station. The grey dotted line symbolizes the average sediment concentration. The 7 cruises are indicated by the black points.

During the sampling period (November 2021 - March 2022), the discharge rate of the Rhône River varied significantly with
monthly fluctuations (Fig. 2). The average river discharge during this period was about 1800 $m^3 s^{-1}$. Daily discharges ranged from 553 $m^3 s^{-1}$, during the low flow period, to 5045 $m^3 s^{-1}$ in January 2023. This maximum river discharge is 3 times larger than the mean discharge experienced during this period and 2 times larger than the other monthly peaks observed in November, December and February. This highest discharge coincides with the maximum of SPM which is as high as 1420 $mg\ L^{-1}$. This high load of SPM is clearly discernible compared to the average SPM of 64 $mg\ L^{-1}$ experienced
within the 5-months duration of the sampling campaign. The other peak in SPM recorded in November 2022 was also relatively small (91 $mg\ L^{-1}$).

**3.2 Porewater composition of DIC, SO₄²⁻ and CH₄ and its comparison to model outputs**

The depth profiles of measured and simulated concentrations of DIC, sulfate, and methane are presented in Fig. 3 for all time points during the winter monitoring. Prior to the flood deposition, porewater sulfate concentrations were constant in the first
10 cm of the sediment with concentration of 31-32 $mM$. Below, $SO_4^{2-}$ concentrations decrease smoothly with depth to 40 cm where no/little sulfate was detected. The model reproduces a strong gradient of sulfate consumption between 10 and 40 cm. This gradient associated with sulfate reduction results in a vertically-integrated sulfate consumption rate obtained by the model of 97 $mmol\ C\ m^{-2}\ d^{-1}$ prior to the deposition event. In contrast, methane was virtually zero in the upper 20 cm of the sediment. From 25 cm downward, methane builds up in the porewater with $CH_4$ rising up to 5 - 6 $mM$ with depth. The
trend in the data, supported by the model, indicates that a linearly diffusing $CH_4$ gradient exists at depth (Fig. 3). At depth, the entire contribution of methanogenesis to organic carbon mineralization as calculated by the model was 9 $mmol\ C\ m^{-2}\ d^{-1}$.

The net product of mineralization pathways is DIC. The measured DIC concentration in the bottom waters was 2.36 $mM$ but gradually increases with depth up to 40 $mM$ at the bottom of the sediment core. This DIC maximum at 40 cm was largely
reproduced by model simulations as a result of organic matter mineralization. In general, DIC and sulfate profiles are symmetrical (negatively correlated) throughout the time series. Before the major flood deposition, the changes of DIC, sulfate and $CH_4$ profiles were limited. However, slight heterogeneity in porewater profile (3 different sampling times) leads to slightly less remarkable agreement between both model and data. Nonetheless, significant degrees of correspondence between the model and data were generally observed with a correlation coefficient higher than 0.8.

After the flood deposition at the end of December 2021, all concentration profiles of chemical species in the sediment porewater showed significant changes. The flood input resulted in the intrusion of sulfate-rich porewater (> 25 $mM$) deep down to 25 cm together with relatively low DIC concentration (< 10 $mM$). This nearly constant sulfate concentration in the data was clearly reproduced by the model (Fig. 3). Below this depth, the profile was similar to the pre-flood situation with a slightly less steep gradient. The strong sulfate consumption between 40 and 75 cm was observed in the data and simulated by
the model. For all three species (DIC, $SO_4^{2-}$ and CH₄), the correlation between observed data and model was significant (r > 0.8). In both cases, a consistent increase to 6 $mM$ for $CH_4$ and to 50 $mM$ for DIC occurred while $SO_4^{2-}$ reached zero at 60 cm. The depth of appearance of $CH_4$ was also shifted from an average position of 30 cm before the flood to a depth of 60 cm after the flood deposition. The maximum in both $CH_4$ and DIC concentrations occured at 70 and 60 cm respectively. In addition, the model reproduced the sulfate-methane transition zone observed in the data. The model was able to obtain a
satisfactory fit generally over the 60 cm depth. Below this depth, the simulated profiles obtained in January 2022 showed a slight deviation from the measured profiles. Below the flood layer, the sediment profiles barely changed in the first month





after the deposition of the flood layer. An excellent agreement between the model and data is observed. Over the longer term, little change in the SMTZ was observed over the two months following flood deposition. The upward diffusion of $CH_4$ was virtually not discernible on the $CH_4$ profiles. Two months after the event, the slow sediment upward shift in $CH_4$ was

still undetectable and the depth of appearance of $CH_4$ was still around 60 cm.

In the upper layer, all solutes were slowly and steadily reorganized one month after deposition. Sulfate was still present down to 60 cm with significant decrease in the top 20 cm, and DIC accumulation in this layer of the sediment was obvious in both data and model results. However, the gradual establishment of a new gradient in this layer begins after that. As of February 2022, two gradients can be seen, one between 19 and 29 cm and the other between 40 and 61 cm.

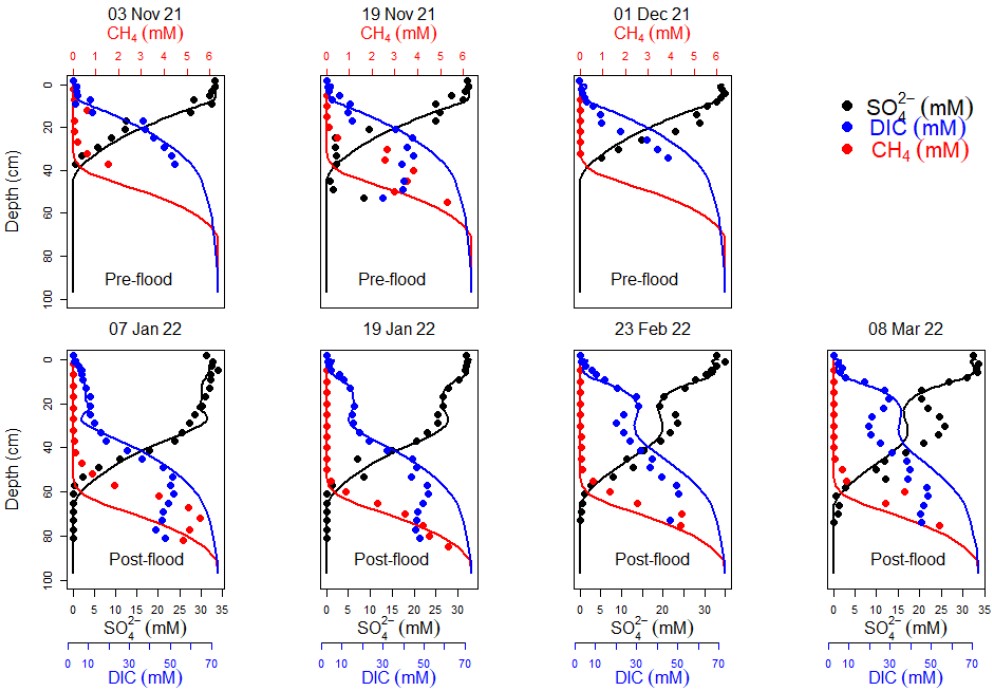

**Figure 3: Vertical distribution of Dissolved Inorganic Carbon (DIC), sulfate ($SO_4^{2-}$) and methane ($CH_4$) concentrations in sediment pore waters. Dots represent the measured porewater data and lines denote the model result.**

### 3.3 Beryllium-7 ($^7Be$)

The vertical distribution of beryllium-7 ($^7Be$) measured after the flood showed significant activity down to 30 cm (Fig. 4). Because of the relatively short half-life of $^7Be$, the presence of $^7Be$ within this active upper zone suggests a recently deposited layer. Higher $^7Be$ activities (up to 58.9 Bq kg-1) are observed in the upper 1.5 cm. Overall, the $^7Be$ activities range between 19 and 54 Bq kg$^{-1}$ on this depth interval and can be considered as relatively homogeneous, considering error bars (2

sigma). No significant $^7Be$ activity is observed below 30 cm. The significant $^7Be$ activities determined as deep as 30 cm thus



likely indicate a recent, instantaneous deposition event. This depth interval likely reflects the thickness of the layer deposited following the flood event.

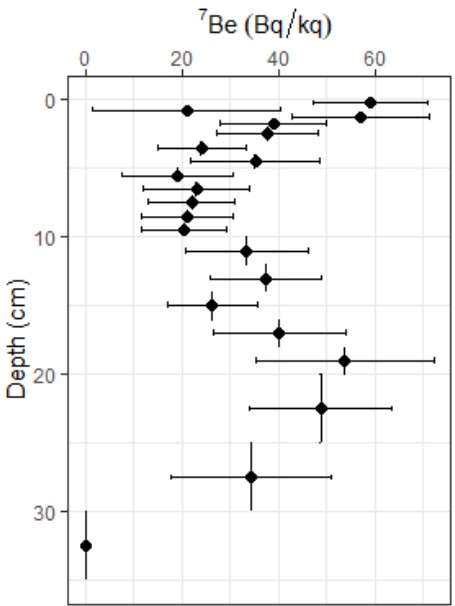

**Figure 4: Vertical distribution of beryllium-7 activities in the sediment core collected after the flood event (07 January 2022), vertical bars represent the thickness of the sediment layer on which analysis were conducted.**

**3.4 Organic carbon content**

The average organic carbon (OC) of sediment cores collected before and after the main flood event is shown Table 3. The
pre-flood sediment has an average OC of $1.3 \pm 0.3$ % dw. The shape of the pre-flood OC profile shows a decrease with depth, starting at $1.8 \pm 0.2$ % dw at the SWI and declining to $0.9 \pm 0.2$ % dw at 22 cm.

In contrast, post-flood OC distribution exhibits a less clear pattern in its variation with depth with overall larger OC content of $1.7 \pm 0.3$%. Like the $^7Be$ profile, two possibly distinct regions are delineated in the OC content profile. In the upper 10 cm of sediment, organic carbon content varies from 2 to 1.3% after the flood deposition. Below, a slight increase in OC
concentration is observed with a subsurface maximum of 1.6%. In the bottom layer, the average OC concentration is larger after the flood ($1.5 \pm 0.3$ %) than before the flood ($0.95 \pm 0.3$%) and is similar to the average OC concentration in the top layer before the flood ($1.6 \pm 0.3$ % dw). As the vertical resolution of the measurement of OC content is coarse in comparison to the beryllium profile, we did not attempt to correlate these two profiles.



| Depth interval (cm) | Pre-flood OC (% dw) | Post-flood OC (% dw) |
|---|---|---|
| 0-1 | 1.8 | 1. 9 |
| 4-5 | 1.5 | 2.0 |
| 9-10 | 1.5 | 1.3 |
| 20-25 | 0.9 | 1.5 |
| 25-30 | 1.0 | 1.6 |

**Table 3: Organic carbon content at selected sediment depth (% dry weight) before and after the flood. Uncertainty of OC is ±**
**0.3%**

## 4 Discussion

Extreme events, such as floods and storms, have measurable impacts on coastal benthic ecosystems near large river mouths and deltas (Tesi et al. 2012; Pastor et al. 2018). The present study provides a temporal picture of the impact of large sediment deposition on sediment biogeochemistry during the winter flooding event of 2021-2022 at a shallow station in the Rhône
River prodelta. Using a combined data-model approach, we describe prominent features of this flood and their implications for carbon cycling in sediments, the evolution of diagenetic pathways and sulfate/methane transformation during early diagenesis.

### 4.1 Disturbance identification, flood and its deposit

The massive deposition of fresh sediments deeply modifies the quantity and quality of the OM and defines the so-called
flood layer. Accurate identification of the flood signature, its thickness and deposition timing which is essential for proper model calibration is challenging. In most instances, the exact specification of when and where the sediment delivered via flood event is permanently deposited on the seafloor is highly uncertain (Tesi et al. 2012) due to possible physical mixing with underneath layer or biomixing (Wheatcroft 1990). Furthermore, while the thickness of the deposited materials during this type of event is an important marker that can be clearly distinguished, it can be smudged by other related events such as
fluctuating deposition-erosion event (Bentley and Nittrouer 2003; Wheatcroft et al. 2006). For large gauged rivers, water discharge that characterizes floods are generally well documented (Zebracki et al. 2015). However, the solid discharge is generally less known due to difficulties in accurately sampling particles during the flood periods.

The average Rhône River water flow was 1470 $m^3\ s^{-1}$ in the winter season of 2021-2022, with short periods of significant higher discharge. There were four periods of increased flow, but only one exceeded the flood threshold of 3000 $m^3 s^{-1}$ at the
end of December. This main winter flood corresponds well with the high concentrations of suspended particulate materials observed in the Rhône River (Fig. 2). Furthermore, Pont et al. (2002) highlighted the non-linear relationship between flows and SPM concentrations which corresponds to large particle discharge at the end of December. Accordingly, a period of time with a single large flood (that is simulated in the model) in the Rhône River prodelta station characterizes this study. This assumption is furthermore supported by the work of Miralles et al. (2005).
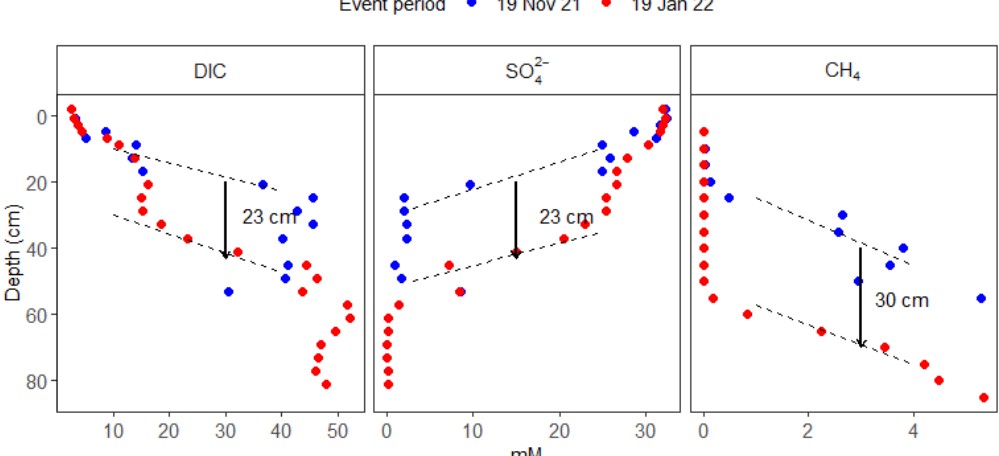

**Figure 5: Concentration profiles of DIC, $SO_4^{2-}$ and $CH_4$ in sediments porewaters from station Z. The dashed lines correspond to the position of the main gradient before (blue dots) and after (red dots) the flood. The arrow symbolizes the shift of this main gradient following the main winter flood.**

In the absence of visual determination of the deposited flood layer, such as variations in the sediment color and texture, we investigated other indicators to evaluate the thickness of the flood layer. The downward shift of the dissolved sulfate gradient ($SO_4^{2-}$) as well as the gradient of $DIC/CH_4$ recorded 15 days after the flood is used to determine the extent of the flood deposition. Our estimate amounts to an average of 25 cm (Fig. 5).

This deposition thickness was validated by analyzing $^7Be$ concentrations measured during this time period, which revealed
significant activities in the first 30 cm of the sediment (Fig. 4). This latter method that allows for the identification of recent sediment deposition of riverine origin has been widely used in other studies documenting flood deposition processes over short time scales (Feng et al. 1999; Palinkas et al. 2005; Wu et al. 2018). Indeed, the $^7Be$ is significantly detected until 30 cm depth (Fig. 4), which indicates newly deposited particles originating from the river down to this particular depth. However, the event layer thicknesses using $^7Be$ can be overestimated in locations where bioturbation activity by benthic fauna is non-
negligible. In the Rhône River prodelta, this is strictly not the case as previous studies have shown that bioturbation rate is low at this location (Pastor et al. 2011a; Pruski et al. 2015) and probably even lower during flood deposition due to habitat disturbance. In general, combining the qualitative assessment of the shift in the post-flood profile relative to the pre-flood, as well as $^7Be$ event-based data, helps in defining our estimate for the deposit thickness. The accurate establishment of this thickness deposit by the flood provides an important constraint to the numerical model and increases its overall skillfulness.

The organic carbon concentration (Table 3) also changes at depth due to the flood deposit. Indeed, the low concentration observed below 25 cm before the flood are refilled by larger OC concentration after the deposit. Furthermore, the new OC concentrations at depth are similar to those found in the top layer before the flood. This clearly indicate a downward shift of the former interface to a depth of 20-25 cm.

**4.2 Transient evolution and mineralization pathways and rates**

The accumulation of large amounts of terrigenous materials in the proximal region of the deltaic depocenter has large implication on the carbon cycle (Hedges and Keil 1995). This routing of carbon to the depocenter sediments results in



substantial organic matter degradation despite acting as accumulation site (Jahnke et al. 1990; Cathalot et al. 2010; Cai 2011; Blair and Aller 2012). The transformation and short-term fate of riverine-OC under episodic events, on the other hand, is largely unknown (Carlin et al. 2021). In the Rhône River prodelta, model estimate of total organic carbon mineralization was

around 148 $mmol\ C\ m^{-2}\ d^{-1}$ before the flood deposition. This estimate is comparable to the total mineralization rate reported in previous studies in the Rhône River prodelta. Under steady state condition, Pastor et al. (2011a) reported a total mineralization rate of 150 $mmol\ C\ m^{-2}\ d^{-1}$ while integrated mineralization rate in Ait Ballagh et al. (2021) averaged around 145 $mmol\ C\ m^{-2}\ d^{-1}$ for the proximal zone of the prodelta. As reported for other coastal systems with organic-rich sediments, anoxic diagenetic pathways involving organoclastic sulfate reduction (OSR) dominated in terms of contribution

to total OC mineralization.

Prior to the flood event, strong sulfate consumption in the surficial sediment was observed in the measured data, as evidenced by a significant decrease in concentration between 10 and 40 cm, accompanied by as significant increase of DIC concentration. DIC accumulation in intermediate sediment layers was also very large for this time period. This pre-flood situation hints at a system under steady state condition. The combined contribution of sulfate reduction and methanogenesis

(> 70% of total mineralization rate) corresponds with values observed in other studies in this shallow region of the prodelta where the anoxic contribution to OC mineralization ranged from 75 % to 89 % (Pastor et al. 2011a; Ait Ballagh et al. 2021).

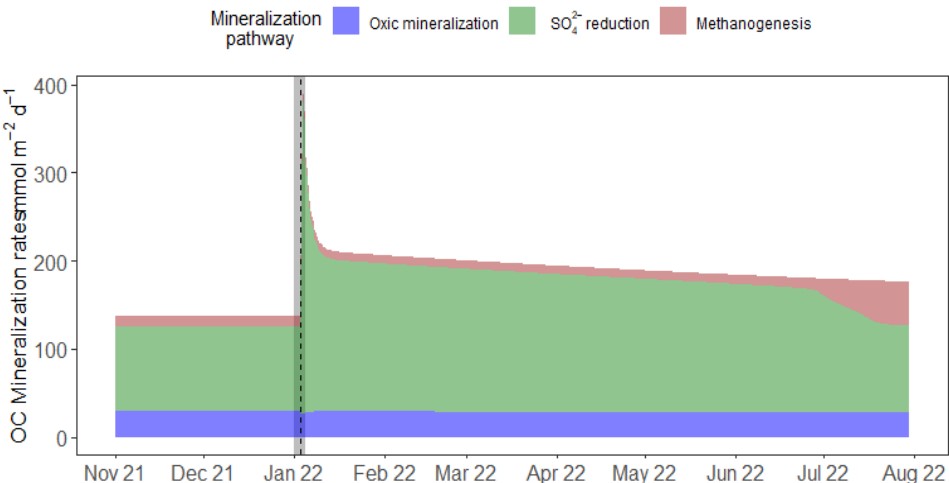

**Figure 6: Vertically integrated rate of organic carbon mineralization in the sediment of the station Z and relative contributions by different pathways. The grey bar and dashed line reflect the date of the main flood (06 January 2022).**

After the flood, the deposition of a thick sediment layer drastically altered the vertical distribution of all profiles with a deeper sulfate penetration, a lower DIC concentration in the top 25 cm of sediment porewaters and a deeper depth of $CH_4$ appearance. The addition of sulfate resulted in intense anoxic-favorable mineralization of degradable OC. As a result, model

calculations suggest an immediate burst followed by an increase of sulfate reduction rates by 75 % (Fig. 6) in comparison to the pre-flood period. As the total rate of OC mineralization increased, the relative contribution of OSR to the total mineralization grows from 65%, before the flood, to 81% after the flood. The relative contribution of methanogenesis to the total OC mineralization rate decreased from 8% to 4% after the flood. At the same time, oxic mineralization which accounts for around 19% of total mineralization before the flood is not modified after the flood due to its very short (daily) relaxation

time (Nmor et al. 2022), and its share in total mineralization decreases to 11%. Thus, immediately after the flood and in the following two months, the OSR is largely favored among the diagenetic pathways in the sediments. This can be related to the





large quantity of sulfate available after the flood deposition which traps sulfate-rich bottom water over the 25 cm added to the sedimentary column, and to its thermodynamically favorable energy yield compared to methanogenesis. These differences in carbon oxidation pathways before and after the flood also reflect the amount and quality of organic matter

deposited in the sediment due to the flood input (Marvin-DiPasquale and Capone 1998; Nmor et al. 2022; Smeaton and Austin 2022). Indeed, these winter floods carry large amounts of metabolizable organic matter originating from terrestrial organic debris or riverine organic matter (Cathalot et al. 2010; Bourgeois et al. 2011; Pozzato et al. 2018) which may trigger intense recycling once deposited in the sediment (Pastor et al. 2018). In a second time period, unfortunately not covered by the data set, model simulations indicate that methane contribution increases following complete sulfate relaxation to its pre-

flood levels 7 months later. The rate of $CH_4$ production by methanogenesis increases, reaching 50 $mmol\ C\ m^{-2}d^{-1}$, i.e. 27% of total mineralization at around 8 months after the event. This secondary increase of methanogenesis needs to be confirmed with new data, it could maintain the long-term relaxation of the system over more than a year for methane, therefore contributing to the accumulation of methane in prodelta sediments (Garcia-Garcia et al. (2006); Nmor et al., In prep)

**4.3 Sulfate-Methane dynamics before and after the flood**

In anoxic sediments, the carbon cycle is tightly coupled to sulfur/methane cycles (Jørgensen and Kasten 2006). The present dataset and model can be used to understand the impact of flood deposition on these coupled cycles. In the case of the sulfur cycle, 90 % of oceanic sulfate reduction takes place in sediments of the continental shelves (Jørgensen 1982; Jørgensen et al. 2019). The two main pathways for sulfate reduction is organoclastic sulfate reduction (OSR) that depends on the lability and

amount of degradable organic matter and anaerobic oxidation of methane (AOM) where methane is anaerobically oxidized to bicarbonate using $SO_4^{2-}$ as electron acceptor by a consortium of microbes including bacteria and archaea (Jørgensen et al. 2019). Although AOM and OSR can coexist, AOM frequently produces a deep sulfate reduction peak different from the shallower maximum of sulfate reduction by carbon oxidation, as it requires considerably lower $SO_4^{2-}$ concentrations. The relative degree of sulfate reduction in both modes regulates the flux of $SO_4^{2-}$ and $CH_4$ across the SWI (Egger et al. 2018).

Because the sedimentary $CH_4$ flux is largely controlled by the rate of AOM, it is critical to understand how $CH_4$ and $SO_4^{2-}$ fluxes are regulated (Dale et al. 2008) especially during flood times and following evolution which disrupts the steady-state control of the $CH_4$ flux. In sediments of the Rhône River prodelta, while bacterial-mediated sulfate reduction is the main oxidation of OC, the quantification of the contribution of anaerobic oxidation of methane is missing from previous studies (Pastor et al. 2011a; Zhuang et al. 2018; Ait Ballagh et al. 2021). In this study, the data-model approach allows us to quantify

the magnitude of the rate of AOM in the sediment.

The depth of maximum AOM before the winter deposition was situated at 35 cm (Fig. 7). The rate of AOM at this depth was 16 $mmol\ m^{-3}\ d^{-1}$. This is higher than values reported in marine sediments from the Skagerrak (5 $mmol\ m^{-3}\ d^{-1}$; Knab et al. (2008)), the Baltic Sea (14 $mmol\ m^{-3}\ d^{-1}$; Treude et al. (2005)), but significantly lower than AOM activities in the Gulf of Mexico or the hydrate ridge off the coast of Oregon (500 $mmol\ m^{-3}\ d^{-1}$; Treude et al. (2003)).





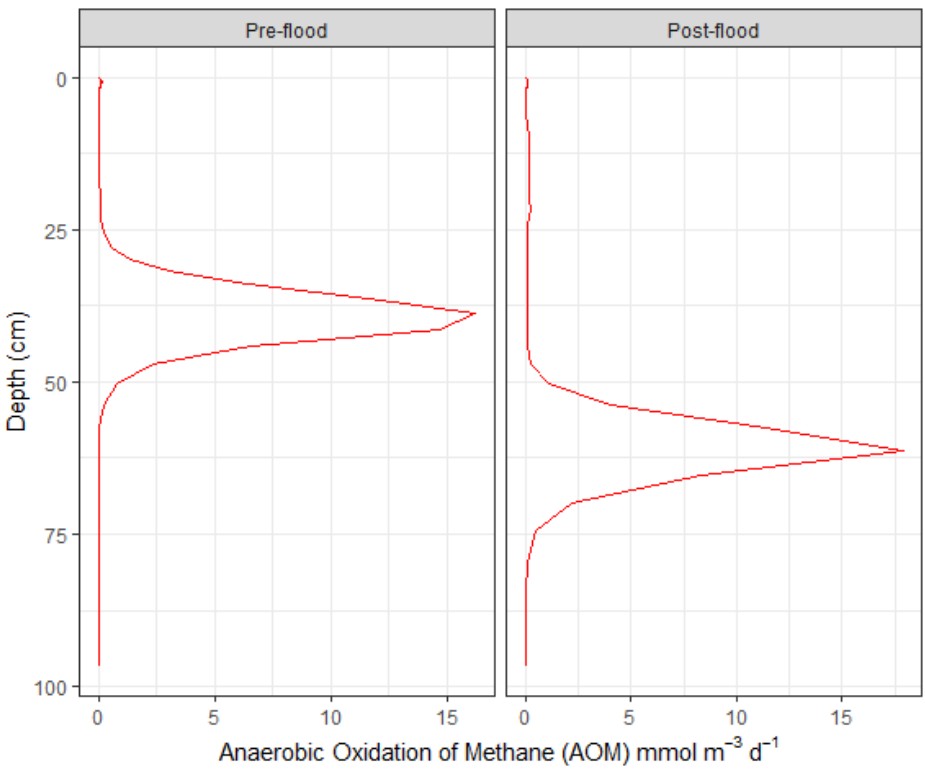

**Figure 7: Vertical distribution of anaerobic oxidation of methane (AOM) for pre-flood and post-flood period.**

After the deposition of the flood layer, the AOM maximum rate remains the same in intensity but is shifted downward in the sediment by 22 cm (Fig. 7). Further cross-examination of the sulfate and methane concentration profiles reveals physical imprint of the flood deposit on the porewater chemical composition. Our data also show that penetration depth of $SO_4^{2-}$ and appearance of methane exhibited a shift downward relative to the pre-flood situation confirming the AOM rate calculation by the model. This generated a downward shift of the sulfate-methane transition zone (SMTZ), defined as the area where

sulfate and $CH_4$ are consumed simultaneously. This SMTZ depth below the seafloor acts as a proxy for $CH_4$ fluxes (Borowski et al. 1999). In general, the presence or absence of externally compressed upward fluid flow (Regnier et al. 2011), the occurrence of localized pockmark where advective transport occurs (Knab et al. 2008) and organic matter load all influence the depth of SMTZ. In our case, the observation of the porewater profiles and the SMTZ suggests a deepening with depth following the introduction of the flood 25 cm layer. Prior to the flood deposition, the SMTZ estimated by the data was

located between 30 and 40 cm whereas the model estimated the precise location of SMTZ at 38 cm. This SMTZ depth shifted to 60 cm after the flood deposition. This vertical shift of the SMTZ in RiOMars system, like the Rhône prodelta, differs from other coastal areas where a shoaling of the SMTZ is experienced as a result of high load of organic matter driven by eutrophication (Crill and Martens 1983). In our case, the deep penetration of bottom water sulfate following the event indicates that $CH_4$ generating processes occurs much deeper. Furthermore, the upward diffusion of the released $CH_4$

(Borowski et al. 1999) is rather slow. This sluggish flux of $CH_4$ to the SMTZ due to slow molecular diffusion of $CH_4$ (Regnier et al. 2011) is linked to the long relaxation timescale associated with processes occurring deep in the sediment (Nmor et al. 2022). Our data provide support to this hypothesis, which shows that the SMTZ in the pre-flood profiles did not change. It is noteworthy that in some other rapidly accumulation setting, increased organic matter load can change the depth





of the SMTZ by bringing it closer to the sediment surface (Crill and Martens 1983; Dale et al. 2019; Myllykangas et al.
2020). The dynamics of this change is unknown and depends on $SO_4^{2-}$ exhaustion by early diagenesis processes. If this is the case, our observation here offers a different view on the role of instantaneous massive flood on sulfur-methane dynamics. This may be due to the low "reactivity" of the organic matter or the short time scale associated with the present study but further investigation of this topic needs to be done to understand the impact of large deposition events.

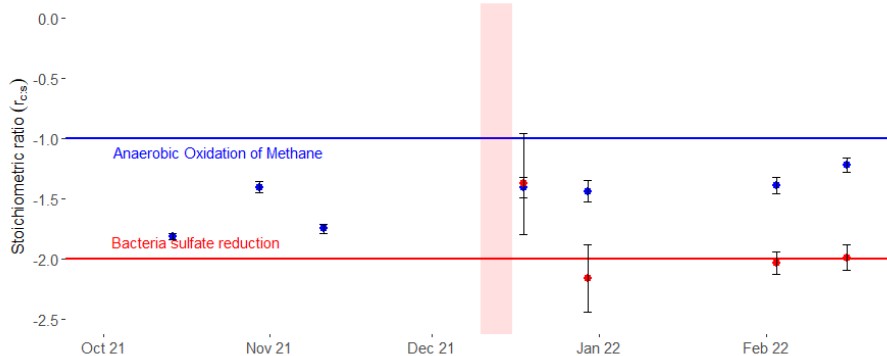

**Figure 8: Temporal variation of DIC:$SO_4^{2-}$ porewater ratio ($r_{c:s}$) calculated at the surface or at depth sediment layer. The red bar indicates the flood period, before the flood the entire core is considered like the surface layer. The blue line indicates the theoretical stoichiometric ratio of the anaerobic oxidation of methane (-1) and the red line indicates the theoretical stoichiometric ratio of the sulfate reduction (-2).**

Quantitative assessment of data based DIC:$SO_4^{2-}$ ratio ($r_{c:s}$) in the sediment cores between pre- and post-flood profiles
reveals a drastic change in the stoichiometric ratios involving sulfate reduction partitioned by depth in the new sediment layer (Fig. 8). Before the winter flood, the $r_{c:s}$ varies between -1.7 and -1.4 with no clear pattern distinguishing the upper and lower zones of the sediment favored by either OSR or AOM. The sudden occurrence of the large sediment deposition triggers a post-flood bifurcation in sulfate reduction delineated by the SMTZ. In the newly deposited layer, the $r_{c:s}$ decreases from -1.8 to -2.0, thus showing a strong tendency toward OSR with time, whereas deeper sediment is AOM-favored with $r_{c:s}$
slowly drifting to -1.1 in February (Fig. 8).

The implication of this event-driven drift between the upper and lower sediment is still unclear. However, the link between the temporal movement of $CH_4$ front and the migration of the AOM activity to changing conditions has been highlighted (Regnier et al. 2011). While the model used here does not explicitly resolve the biomass involved in the reactions (Dale et al. 2007) or consider the impact of bioenergetics (Dale et al. 2006), we show that a shift in the SMTZ is correlated with the
depth of maximum AOM rate before and after the deposition (Fig. 7). Since $SO_4^{2-}$ and $CH_4$ data are correctly reproduced by the model, the depth of the maximum AOM rate is thus essentially controlled by methane availability. This deepening of the AOM maximum suggests that in the advent of flood deposition, the AOM traps more efficiently the upward flux of methane. It has been suggested that the advent of shallower SMTZ would provide larger chance of methane escape from the sediment to the overlying water and, ultimately, to the atmosphere (Borges and Abril 2011). Thus, the occurrence of this large
deposition and the associated downward shift of SMTZ could increase the efficiency of the $CH_4$ trapping in sediments.

### 4.4 Flood induced fluxes and link to carbonate chemistry

Because of their high load of mineralizable organic matter, coastal sediment represents an important source of $CO_2$ to the coastal ocean and to the atmosphere (Egger et al. 2016). Changes in the intensity of various mineralization processes in



response to flood deposition raise concerns about the consequences on fluxes of dissolved inorganic carbon at the sediment-
water interface. This flux may have a broader impact on benthic-pelagic biogeochemistry, such as ocean acidification ($CO_2$)
of the coastal waters. Current estimate of solutes release does not explicitly account for these event-driven sedimentations
which might have different geochemical properties depending on the type of flood (Cathalot et al. 2010; Pruski et al. 2015).
For example, our results show that the event deposits have higher % OC values and drive larger mineralization rates (Fig. 6)
which result in substantial change of the sediment interstitial composition and possibly fluxes.

A remarkable change in the benthic exchange across the sediment-water interface was observed for DIC (Fig. 9). Before the
flood deposition, the DIC flux out of the sediment amounts to 101 $mmol\ m^{-2}\ d^{-1}$. This calculated DIC efflux is larger than
previous data based estimate but remains in the same order of magnitude as flux estimate reported in this proximal zone (18 -
78 $mmol\ m^{-2}\ d^{-1}$; Rassmann et al., 2020) as well as other river deltas: Mississippi River delta sediment (36 - 53
$mmol\ m^{-2}\ d^{-1}$; Rowe et al., 2002), Fly River delta (35-42 $mmol\ m^{-2}\ d^{-1}$; Aller et al., 2008) and Guadalquivir River
estuary (36-46 $mmol\ m^{-2}\ d^{-1}$; Ferrón et al., 2009). After the flood deposition, the model estimates of DIC benthic flux
decreased from 101 to 55 $mmol\ m^{-2}\ d^{-1}$ in response to the new input. This was largely related to the large decrease of the
DIC gradient in porewaters after the flood (Fig. 3) and represents a 45% reduction in DIC flux out of the sediment shortly
after flood deposition. The reduced DIC flux rapidly resumed to the previous situation after a week of lower fluxes and
stabilized to a value a few percent above the initial value. Yet, the production of DIC in the sediment column had increased
by 43% due to the sudden increase of OC recycling activity following the introduction of fresh organic carbon contained in
the flood deposit (Fig. 9). The initial decrease of the flux of DIC was followed by a slight increase of about 4% and then a
stabilization at almost the same initial value as before the flood indicating that most of the DIC produced by the flood
deposit is stored in the sediment porewaters. This is obvious from the DIC profile (Fig. 3) which clearly indicates an
accumulation of DIC in porewaters after the flood along the measurement period (from January to March).

This change of DIC flux in response to the abrupt introduction of flood-driven deposit can have an impact on the
contribution of coastal sediments to the release of $CO_2$ to the coastal zone and potentially later to the atmosphere. The extent
of this gas exchange is determined by several factors, including the DIC/Total Alkalinity (TA) flux ratio (Andersson and
Mackenzie, 2012). In the Rhône prodelta sediments, the total alkalinity flux ranges from 14 to 74 $mmol\ m^{-2}\ d^{-1}$ thus
acting as an efficient counteracting mechanism controlling DIC fluxes to the overlying water (Rassmann et al. 2020). As
most of the increase of DIC production raised from organoclastic sulfate-reduction which has a DIC/TA production ratio of 1
(Rassmann et al. 2020), the flux of alkalinity out of the sediment would probably follow the DIC flux, therefore bringing
little changes to the DIC/TA ratio in the coastal bottom waters (Hu and Cai 2011). However, other contributors to sediment
alkalinity such as calcium carbonate dissolution as well as potential coupling processes involving $FeS$ and $FeS_2$ burial might
well affect the alkalinity during relaxation of the system after the flood (Nmor et al., in prep). This is likely the case in the
Rhône River prodelta sediment where substantial pyrite burial at depth has been reported (Rassmann et al. 2020). As direct
measurements of DIC and total alkalinity flux at this winter flood are unavailable, and porewater iron and sulfide were not
monitored during the time series, we can only speculate with the model results. In any case, the magnitude of the DIC flux
decrease and its internal storage in surface sediment porewaters highlight the need to better study this phenomenon and
provide better constraints on their contribution to coastal carbon cycle.





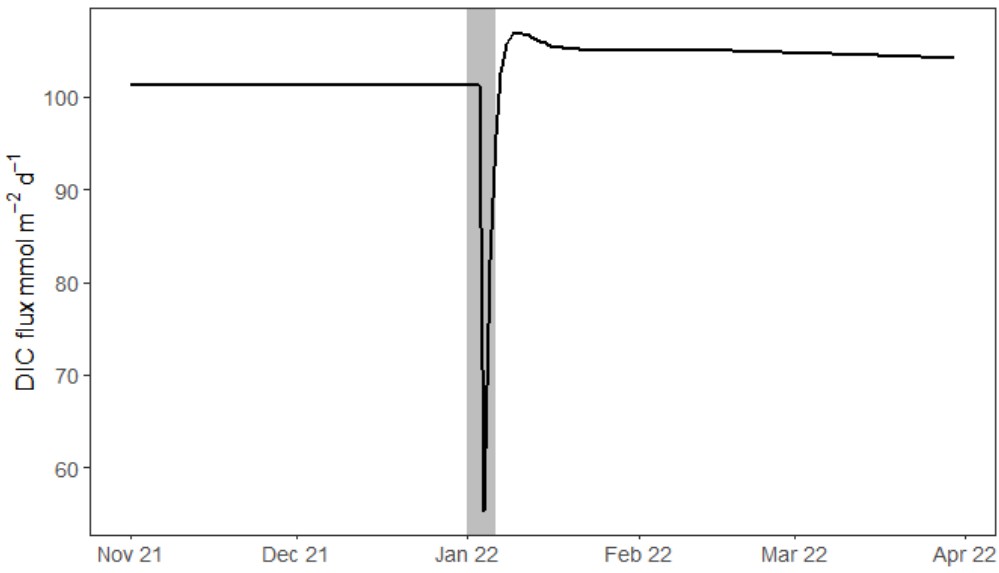

**Figure 9: Flux of DIC across the sediment-water interface. Positive flux is directed from the sediment to the overlying bottom water. The grey bar reflects the date of the main flood (06/01/2022).**

## 5 Conclusion

Extreme flood deposition events produce transient dynamics in biogeochemical processes in coastal marine environment. In this paper, we documented the temporal features of porewater short-term response over 2 months to organic matter flood input at a station located in the Rhône prodelta. Using a data-model approach, we showed that the introduction of this new layer of OM input from flood deposition can alter the porewater profile of $SO_4^{2-}$, DIC and methane. Although the model incorporates some simplifying assumptions, FESDIA is able to reproduce accurately the measured concentration depth profiles, including time and space variations. This reflects the capability of the model to capture non-steady state dynamics driven by abruptly changing boundary conditions.

Our modelling results indicate that large amounts of sediment can also trigger intense biogeochemical processes with stimulation of sulfate reduction and immediate decline of DIC flux out of the sediment. The internal storage of DIC in porewaters indicates a relative decoupling of sediment organic matter mineralization and fluxes to the water column. By considering the measured porewater profiles and the reactions stoichiometry, we showed that massive deposition of sediment can results in the disconnect between anaerobic oxidation of methane and organoclastic pathway for sulfate reduction in the sediment. This decoupling of AOM and OSR implies an increase in the efficiency of the sediment capacity to trap incoming flux of methane from below. The immediate consequence of the changes in the porewater chemistry and processes following these events highlights their importance in the short to medium term response and system functioning in the respective biogeochemical cycle. On a broader scale, the long-term fate of such events on sediment biogeochemical processes will require better and more continuous field monitoring to help future model development addressing the biogeochemical consequences of these flood events.

*Code and data availability* All data used in this study are available through SEANOE (https://doi.org/10.17882/96514) (Ferreira et al., 2023). The model version used to produce the results used in this paper is archived on Zenodo (https://doi.org/10.5281/zenodo.6369288) (Nmor et al., 2022).



*Author contributions.*: EF, SN, EV, BB, BL, CG and CR conceptualized the study. EF, SN, BB, ER, GM, PvB and CR contributed to the data curation. EF, SN, CG, PvB performed the formal analysis. CR acquired the funds. EF, SN, EV, BB, BL, ER, CG, PvB and CR made the investigation. EF, SN, EV, BB, BL and CR contributed to the methodology. BB, CG and CR are behind the project administration and the supervision of the research planning. SN, EV and CR contributed to the software development. EF and SN contributed equally to the visualization and the writing of the original draft preparation. All authors contributed to the writing review, editing and validation.

*Competing interests.* The authors declare that they have no conflict of interest.

*Disclaimer. Publisher's note:* Copernicus Publications remains neutral with regard to jurisdictional claims in published maps and institutional affiliations

*Acknowledgements.* The authors would like to acknowledge Nolwenn Verpy, Laure Papillon and Deny Malengros for help collecting samples, Caroline Gauthier for help with OC analysis, Marc Souhaut, and Thomas Zambardi for help with 7-Berylium analysis at the LAFARA underground laboratory. We would like to thank Hugo Lepage for discussing river flows and helping us obtain the SPM data and also the Observatoire des Sédiments du Rhône staff for the SPM results. Finally, we would also like to thank the various members of the RV Antédon II crew who were present on several campaigns. The map in this study was created with the ODV Software Schlitzer, Reiner, Ocean Data View, https://odv.awi.de, 2022.

*Financial support.* This work was supported by a grant from the special call on estuaries from French National programme EC2CO under the name "DeltaRhone" and the French government grant managed by the Agence Nationale de la Recherche under the France 2030 program, RiOMar reference ANR 22 POCE 0006

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
