# Peer review of "Characterization of the benthic biogeochemical dynamics after flood events in the Rhône River prodelta: A data-model approach"

_Biogeosciences, 2023_

## Referee Comment (RC2)

Biogeosciences Open Access

Discussions

EGU

[referee-annotated manuscript omitted]

---

## Author Response (AR1)

**Author's response**

*Dear Editor,*

*Below, you will find our updated submission in response to the feedback provided by the reviewers. Additionally, we have included a version of the manuscript with tracked changes, which clearly indicates all insertions and deletions made in response to each point raised by the reviewers. We'd like to thank you for your help with the submission process, and we'd like to thank the two reviewers for their corrections and comments. We believe that these revisions effectively address the concerns raised by the reviewers.*

*Best regards,*

*Eva Ferreira, on behalf of all authors.*

**REVISIONS IN RESPONSE TO COMMENTS BY ANONYMOUS REFEREE #1**

**Main comments :**

- **RW1 :** My major concern likely originates from a personal misunderstanding of the FESDIA model concept. Still, this needs to be clarified or some statements require reconsideration. As I understand, in FESDIA (as in OMEXDIA) the total carbon mineralization rate is unaffected by the availability of oxidants. As Eq. 1 in S. I. Nmor et al.: FESDIA (v1.0), 2022 depicts, each OM fraction has a fixed degradation rate. The total mineralization rate changes with depth following the relative proportion of the labile and semi-labile OM fraction, but not because one or the other of the successive oxidants is more or less abundant. What changes with oxidants' availability is the process by which OM is mineralized, hence the products of these reactions, but not the total carbon mineralization rate itself. Either I'm wrong, or the following statements need to be reconsidered/reformulated/clarified, in line with this important assumption of the FESDIA concept.

**Authors :** The reviewer is correct with the assertion that the total organic carbon (OC) mineralization rate changes with depth and time following the concentration of the labile and semi-labile OC fractions and is directly decoupled from the oxidant available i.e. $C_{miner} = r * OC$ (where r is the rate constant and OC the organic carbon concentration for labile or semi-labile OC). This parameterization indeed indicates that the rate of carbon mineralization is independent of the electron acceptors. The reviewer is right in stating that the rate of metabolic activity for the oxidant and its by-products depends not only on the degradability of OM but also on the availability of the oxidants utilized as expressed by the limiting and inhibition functions (Eq 3 & 4 Nmor et al 2022). Using that expression in Eq 1 with the combined Eq 3 and 4 in Nmor et al 2022 and similar in Soetaert et al 1996a allows for a single equation of total carbon mineralization while partitioning of the carbon mineralization to the various pathways. To that end, we will reformulate the different statements suggested by the reviewer to provide clarity on the model concept and results

- **RW1, L24-26:** "(rephrased) the flood event increases total carbon mineralization rate by increasing sulfate reduction contribution to this total"? ( the flood increases total

mineralization AND sulfate contribution, but the first is not a consequence of the second).

**Authors :** The sentence will be restructured to remove the causal-correlation of total mineralization and sulfate reduction :

*"The use of an early diagenetic model (FESDIA) to calculate biogeochemical reaction rates and fluxes revealed that this type of flood event can increase the total organic carbon mineralization rate in the sediment by 75% a few days after deposition. In this period, sulfate reduction is the main process contributing to the increase in total mineralization relative to non-flood deposition."*

- **RW1, L176**: "This decay is modelled as a first order rate law and is dependent on the limitations of specific oxidants" (to be precised. The rate of the decay is not dependent, the result or product of the decay is.)

**Authors :** L176 will be paraphrased to buttress the explanation given above and will be clarify thus as :

*"The decay is modelled as a first-order rate law and the contribution of the different mineralization pathways to total OC mineralization is determined by the limitation and inhibition of the different primary oxidants in the sediment."*

- **RW1, L396:** "The addition of sulfate resulted in intense anoxic-favorable mineralization of degradable OC" (Again, the addition of sulfate increases the role of sulfate, but not mineralization intensity. What changes mineralization intensity however, and this could be stated more explicitly although discussed in L381-383 & L470, is the addition of OMfast).

**Authors :** L396 will be removed as it is redundant without changing the meaning and context of the paragraph block.

**Minor comments/Typos**

- **RW1, L17:** Not only the short-term impact is unknown. Considering the dynamic RioMARS systems and their response to land discharges, it seems to me that the long-term effect of highly intermittent discharge, instead of the smoother forcing signal often considered by default, remains elusive to present-day science as well. Clearly, the authors are well aware of this and remain cautious not to go beyond their results. But I think that, in some places of the manuscript, this aspect - that the default steady-state approach is insufficient where relaxation time is larger than the return period of disruptive events - may be stressed more explicitly.

**Authors :** We acknowledge that the long-term effect of flood deposition is important, not only the short-term. This was already indicated in the Conclusion section (line 506). In this paper, the data covered a limited duration of 2.5 months after the flood and we decided to cover the short-term response only in this data-model approach. We will add a second sentence in the conclusion stressing the need for long-term studies including the interactions between flood deposition.

*"On a longer timescale, the complete relaxation time potentially longer than the return time of disturbing events may also influence benthic biogeochemical fluxes. It is urgent to address the issue of multi-seasonal evolution of the sedimentary system after flood deposition together with the potential interaction between successive flood deposits. Clearly, the long-term fate [...]"*

- **RW1, L44:** a_naerobic -> anaerobic

**Authors** : This will be corrected on the revised text.

- **RW1 :** L46: AS -> As

**Authors :** This will be corrected on the revised text.

- **RW1 :** L115: Since it only appears here, one could maybe spell "Cesium" explicitly instead of "Cs"?

**Authors :** This will be added to the revised text.

- **RW1 :** L235-236: In general, I guess that the characterization of the newly deposited material bears a strong impact on the simulation result. These lines suggest a single enrichment factor, whereas table 2 list in fact 6 alpha factors, two of which required adjustments to fit the observations. I think this is an important aspect that would deserve further discussion.

**Authors :** The reviewer is right: there are multiple enrichment factors that are used to mimic the input of the different compounds during the flood layer deposition (see Table 2). The lines that suggest a single enrichment factor will be changed to "Depending on the nature of the flood, it can be enriched or depleted in reactive compounds for the two pools of organic matter ($Cfastorg$, $Corgslow$), of manganese oxides ($MnO2A$ and $MnO2B$) and amorphous iron ($FeOOHA$ and $FeOOHB$). This is translated in the model by the enrichment factors ($\alpha$), specific to each type of compounds which were set to the values reported in Table 2."

- **RW1, L189**: Remove "that remains" and "subsequently"

**Authors :** This will be removed on the revised text.

- **RW1,L190**: Reaction 3 -> (Reaction 3)

**Authors :** This will be corrected on the revised text.

- **RW1, L195**: deeper -> deep

**Authors :** This will be corrected on the revised text.

- **RW1, L197**: occurs -> occur

**Authors :** This will be corrected on the revised text.

- **RW1, L198:** "into either free gas" ?

**Authors** : This will be corrected on the revised text as : "into free gas"

- **RW1, L200:** Is a verb missing in the sentence?

**Authors** : This sentence will be rewritten as:

*"[..]where $CH_{4equil}$ is the equilibrium concentration for which observed/simulated methane goes into the hydrate or gas phases."*

- **RW1,L211:** Stop the sentence after "AOM reaction ate", and restart a new sentence with "As this"

**Authors** : This will be corrected on the revised text.

- **RW1, L221:** "following the obtained data" -> "according to observations".

**Authors :** This will be corrected on the revised text.

- **RW1, L225-236:** Please describe how solute species are handled during the deposition event. This is important for the results (eg. discussion on high sulfate concentration in porewaters after deposition).

**Authors :** We omitted that part because it was already discussed in Nmor et al 2022. However, we will add a sentence concerning solid and solutes change due to the event.

*"The deposition of flood materials was simulated in similar manner as described in Nmor et al. (2022). Here, we imposed a singular flood scenario with a thickness of 25 cm[…] porewater solute distribution and strengthened by beryllium-7 data collected after particle settling (see Section 3). During the time of the event, the model assumed the solute concentration within the newly deposited layer is homogenous and it resets to the bottom water concentration".*

- **RW1, L251:** Is it " a gradient of sulfate concentration" or simply "sulfate consumption" rather than "a gradient of sulfate consumption"?

**Authors :** This particular sentence will be rewritten as:

*"The model reproduces a strong gradient of decreasing sulfate concentrations between 10 and 40 cm."*

- **RW1, L255:** rephrase "a linearly diffusing gradient"

**Authors :** The sentence will be rewritten as: "The trend in the data, supported by the model, indicates that a $CH_4$ gradient exists at depth."

- **RW1, L255:** no need to repeat (Fig. 3). This is true for the rest of the section.

**Authors :** This will be corrected for all the section on the revised text.

- **RW1, L259:** clarify "largely reproduced by the model"

**Authors :** The sentence will be rephrased as:

*"The DIC maximum at 40 cm was well reproduced by the model simulations due to the accumulation of [...]"*

- **RW1 :** L279: remove "sediment"

**Authors :** This will be removed to the revised text.

- **RW1, L285:** Would it make sense to switch the order of Sect 3.3 and 3.2 ? 3.3 is independent from 3.2, while 3.2 has several references to the "flood layer" which is characterized in 3.3.

**Authors :** We understand the proposal to switch sections 3.2 with 3.3, and why it is being proposed. Nevertheless, this ordering was intentional. Section 3.2 with Figure 3 is the heart of our work and was therefore placed first. The observation of the shift in concentration gradient observed in Fig. 3 from pre-flood to post-flood shows a 25cm layer of deposition. Section 3.3, independent of section 3.2, supports the observations that are made in this previous session. The $^7$Be results then support this analysis.

We conclude that presenting this primary result first before other supporting data presented in our study is a better sequence for public readership.

- **RW1, L287:** Please provide an indicative number for 7Be half-life.

**Authors :** This will be added to the revised text.

- **RW1, L294:** Add: organic matter "content"

**Authors :** This will be added to the revised text.

- **RW1, L315:** Add: Thickness, deposition timing, "and composition" (cf. discussion on alphaXXX parameters).

**Authors :** This will be added to the revised text.

- **RW1, L349:** Transient evolution OF mineralization pathways and rates.

**Authors :** This will be corrected to the revised text.

- **RW1 :** L419: the 25cm flood layer.

**Authors :** This will be corrected to the revised text.

- **RW1,L502:** "the disconnect" -> a disconnection

**Authors :** This will be corrected to the revised text.

- **RW1, L503** rephrase "incoming"

**Authors :** This will be rephrased to the revised text.

**Table 2:**

- **RW1 :** Pay attention to format units correctly (eg. with exponents as exponents).

**Authors :** The correct format of the units will be revised in the main text.

- **RW1 :** Values for rMnFe and RAOM are given with different scientific number conventions.

**Authors :** The values of rMnFe and rAOM will be formatted to a common scientific number convention (i.e. exponent)

- **RW1 :** Figure 3: Is it possible to add error bars on the observation points?

**Authors :** The error bar in the observation points have been added to the figure.

[Figure]

**Table 3**

- **RW1 :->** As a figure (with error bars)?

**Authors :** We prefer to keep the Table as it is (with the error inserted in the caption) instead of a Figure that is bringing no additional values to these sparse data.

- **RW1 :** Why are model values not indicated?

**Authors :** We have calculated the averages of the model outputs for each period, which we will be added to the revised text.

**REVISIONS IN RESPONSE TO COMMENTS\* BY ANNIET LAVERMAN**

**\* The comments of the second reviewer, Anniet Laverman, were largely made on the document attached to the review, to which you can refer if necessary (https://doi.org/10.5194/bg-2023-166-RC2))**

- **ANNIET LAVERMAN :** In Table 2 the abbreviations are not explained, please add information on that.

**Authors :** The abbreviations of Fig. 2 will be added in the caption (FDET and SDET)

- **ANNIET LAVERMAN :** It should somewhere be stated that the test was done (figure 5) to prove the addition of the organic matter rich layer for the 19 November sampling. Were the two other sampling periods also tested? Something should be added justifying the use of only one sampling period. This will not change the major results, overall the justification of the organic carbon layer is clear.

**Authors :** We understand the reviewer's comment as a question on testing the post-flood changes in porewater distribution versus different pre-flood conditions. It is clear that no deposition of organic-rich layer was affected on 19th nov, as the deposition arised on dec 31st. As can be seen on Fig. 3, the three datasets before the flood (3 Nov, 19 nov and 1st Dec 2021) are very similar. Using one or the other for the calculation of the deposited thickness would therefore be similar. The choice of these two sampling periods in Fig. 5 (19th nov and 19th Jan) is shown to illustrate the effect of the deposit layer. The result would be very similar for other periods. We will add a sentence to clarify that point :

*"In this example, we used the difference between 19 November 2021 and 19 January 2022. Using other pre-flood profiles would provide a similar estimate of the deposition thickness, as pre-flood profiles are very comparable (Fig. 3)."*

- **ANNIET LAVERMAN :** Please check the tenses, in the discussion sometimes it is present, sometimes past tense: example L370 "grows" , on the next line "increased" … and L373, 374 "accounts" and "is not modified" – these are just some examples.

**Authors :** The tenses will be homogenized in the final draft.

- **ANNIET LAVERMAN,** L24-26 "sulfate reduction contribution" .

**Authors :** Sentence modified considering the comments of the two reviewers.

*"The use of an early diagenetic model (FESDIA) to calculate biogeochemical reaction rates and fluxes revealed that this type of flooding event can increase the total organic carbon mineralization rate in the sediment by 75% a few days after deposition. In this period, sulfate reduction is the main process contributing to the increase in total mineralization relative to non-flood deposition."*

- **ANNIET LAVERMAN,** L33 "This depth-wise bifurcation of both pathways of sulfate reduction in the sediment column is clearly related to the deepening of the sulfate-methane transition zone (SMTZ) by 25 cm after the flood deposition" rephrase please.

**Authors :** This sentence will be rephrased on the revised text.

"*The bifurcation depth of sulfate reduction pathways, i.e. the sulfate-methane transition zone (SMTZ), is shifted deeper by 25 cm in the sediment column following the flood deposition.*"

- **ANNIET LAVERMAN,** L46 "AS: As"

**Authors :** Correction will be made

- **ANNIET LAVERMAN,** L85 "is :was"

**Authors :** Correction will be made

- **ANNIET LAVERMAN,** L134 "indicate why the KOH is used ? "

**Authors :** We use KOH for methane preservation based on the protocol developed by Magen et al. in 2014 (doi : 10.4319/lom.2014.12.637) . The KOH solution inhibits microbial activity it also neutralizes acidic compound that may be present to maintain the stability of methane concentration and prevent chemical reactions.

- **ANNIET LAVERMAN,** L193 "what is high apparent sediment deposition ?"

**Authors :** We chose to remove the word "apparent", as it was redundant with evidence.

- **ANNIET LAVERMAN,** L236 explain $\alpha$

**Authors :** It is explained in text:

"*As described in Nmor et al. (2022), the deposited flood layer can have a different particulate composition than the preexisting sediment. Depending on the nature of the flood, it can be enriched or depleted in reactive compounds (two pools of organic matter ($C_{fast}^{org}$, $C_{org}$ $slow$), of manganese ($MnO2A$ and $MnO2B$) and amorphous iron ($FeOOHA$ and $FeOOHB$)) which is translated in the model by the enrichment factor ($\alpha$).*"

- **ANNIET LAVERMAN :** Figure 2 : "what is SORA station ?"

**Authors :** SORA is the name given by IRSN and the Rhône Mediterranée et Corse water agency to the reference station where measurements are taken as part of Rhône monitoring. It stands for "Station Observatoire du Rhône à Arles". This is the station closest to the river mouth, allowing us to track the river's inflow (and SPM) before the flows in the pro-delta. We added the location of the SORA station in the caption: "*… at SORA station in Arles*".

- **ANNIET LAVERMAN,** L307 remove the sentence "Extreme events, such as floods and storms, have measurable impacts on coastal benthic ecosystems near large river mouths and deltas (Tesi et al. 2012; Pastor et al. 2018)."

**Authors :** We prefer to keep this general sentence at the beginning of the Discussion

- **ANNIET LAVERMAN,** L322 "particles, please rephrase not clear"

**Authors :** This will be rephrased on the revised text as :

*"[…] challenges in accurately sampling sediment during the flood periods."*

- **ANNIET LAVERMAN,** L340 "remove strictly"

**Authors :** Correction made

- **ANNIET LAVERMAN,** L345 "remove indeed"

**Authors :**Correction made

- **ANNIET LAVERMAN,** L366 "what about the other similar areas ? any data on that ?"

**Authors :**In one of the references Pastor et al. (2011a), the authors compile a large data set (Figure 6) relating the percentage of anoxic mineralization to the OC fluxes in 9 other sites. In the revised version we will add "and references therein" with the citations.

- **ANNIET LAVERMAN,** L388 "remove brackets"

**Authors :**Correction made

- **ANNIET LAVERMAN,** L396 correct the reference, this is not the correct to refer to AOM.

**Authors :** Correction made, we will change Jørgensen et al., 2019 to Boetius et al., 2000.

- **ANNIET LAVERMAN,** L398 "why does it require lower sulfate concentrations? please add a reference":

**Authors :**AOM is generally located in the SMTZ where the sulfate concentrations are low. We will change the sentence L398 to :

*"Although AOM and OSR can coexist, AOM frequently produces a deep sulfate reduction peak different from the shallower maximum of sulfate reduction by carbon oxidation, as it essentially occurs in the SMTZ with low $SO_4^{2-}$ concentrations (Regnier et al., 2011)."*

- **ANNIET LAVERMAN,** L400 "don't start a sentence of a new paragraph with "because".

**Authors :**This will be corrected on the revised text.

- **ANNIET LAVERMAN,** L403 "oxidation :oxidation process"

**Authors :**This will be corrected on the revised text.

- **ANNIET LAVERMAN,** L408 "remove brackets".

**Authors :**This will be corrected on the revised text.

- **ANNIET LAVERMAN,** L419 "25 cm flood layer"

**Authors :**This will be corrected on the revised text.

- **ANNIET LAVERMAN,** L441 tense use

**Authors :**The tenses will be homogenized in the draft.

- **ANNIET LAVERMAN,** L452 "don't start a sentence of a new paragraph with "because".

**Authors :**This will be corrected on the revised text.

- **ANNIET LAVERMAN,** L456 "A current estimate or current estimates"

**Authors :**This will be corrected on the revised text.

- **ANNIET LAVERMAN,** L469 add "of" "

**Authors :**This will be corrected on the revised text.

- **ANNIET LAVERMAN,** L484 is " in prep" a valid reference ?

**Authors :** This will be corrected on the revised text.

- **ANNIET LAVERMAN,** L486 "at"

**Authors :**This will be corrected on the revised text.